# Validation of the Italian Version of the Behavioral Inhibition Questionnaire (BIQ) for Preschool Children

**DOI:** 10.3390/ijerph18115522

**Published:** 2021-05-21

**Authors:** Francesca Agostini, Mariagrazia Benassi, Marianna Minelli, Luca Mandolesi, Sara Giovagnoli, Erica Neri

**Affiliations:** Department of Psychology “Renzo Canestrari”, University of Bologna, 40127 Bologna, Italy; f.agostini@unibo.it (F.A.); marianna.minelli2@unibo.it (M.M.); luca-mandolesi@libero.it (L.M.); sara.giovagnoli@unibo.it (S.G.); erica.neri4@unibo.it (E.N.)

**Keywords:** temperament, behavioral inhibition, anxiety, preschool children, Italian sample, psychometrics, behavioral inhibition questionnaire/BIQ

## Abstract

Behavioral Inhibition (BI) is a temperamental trait characterized by fear and wariness in reaction to new and unfamiliar stimuli, both social and non-social. BI has been recognized as possible forerunner of anxiety disorders, especially social anxiety and phobia; therefore, its assessment is clinically relevant. The present study aimed to examine the psychometric properties of the Italian adaptation of the Behavioral Inhibition Questionnaire (BIQ), which measures BI in preschool children. The BIQ was completed by 417 Italian parents (230 mothers, 187 fathers) of 270 preschoolers aged 3–5. Confirmatory factor analysis showed a good internal validity: the factorial structure was corresponding to the original six-factor version. Results showed excellent internal consistency, significant item-total correlations, good inter-rater reliability, convergent validity (by correlating the BIQ with the Italian Questionnaires of Temperament-QUIT, the Anxiety-Shy Conner’s Scale and the Laboratory Temperament Assessment Battery) and discriminant validity (i.e., no correlation with Conners’ ADHD scale). Significant correlations emerged between BI indexes and total BIQ scores of parents and maternal (but not paternal) versions of the questionnaire. Altogether, the results are promising and consistent with previous validation studies, suggesting the BIQ as a reliable and valid measure for evaluating parents’ perception of BI in Italian preschoolers.

## 1. Introduction

The psychological construct of Behavioral Inhibition to the Unfamiliar (BI) [1] refers to an innate temperamental trait that exists in 10–15% of Caucasian population and can be observed as early as 14 months of a child’s age [2]. BI has been originally described as the *“the child’s early initial behavioral reactions to unfamiliar people, objects, and contexts, or challenging situations”* (p. 53 in [3]), and concerns a general tendency to be unusually shy and to respond with negative reactivity, fear or withdrawal in unfamiliar situations [4]. The BI trait is in fact characterized by both emotional aspects, such as fear and distress in presence of novelty [5,6], and behavioral aspects, like close proximity to the caregiver, decrease of smiles and vocalizations, freezing, avoidance and reticence in presence of new or unfamiliar stimuli [5,7].

The definition of the temperamental trait of BI requires a distinction from other similar constructs. First, BI should be distinguished from the construct of effortful control, and especially from *inhibitory control*, that is the ability to voluntarily inhibit impulsive behaviors to promote more adequate responses according to the characteristics of the context [8,9,10], while BI refers to an involuntary process [11]. Second, with regards to *fear* and *anxiety*, BI is obviously a related-fear construct, where fear can be defined as an emotional adaptive response since infancy to unfamiliar stimuli. Fear is expressed according to individual differences and environmental characteristics and only high fear is associated to BI as a specific temperamental disposition; BI, in turn, has been recognized as a risk factor for the occurrence of later anxiety [12]. BI also differs from *shyness*, which represents wariness and reticence but only refers to socio-evaluative settings; on the contrary, BI refers to a general avoidance of novelty, independently of evaluative situations. Specifically, the BI trait is characterized by the presence of two core components: the response to social stimuli (e.g., stranger adults or peers), and the reaction to nonsocial stimuli (such as new objects, food, physical activity) with risk of injury and uncertainty [13]. The dependent or relatively independent relation between social and nonsocial aspects of BI is still an open debate in literature but, according to the position of Kagan, Snidman and Arcus (1998), these two components would be interdependent from each other. Kagan et al. (1998) highlighted that a young child classified as inhibited might, with experience, diminish the initial reticence with strangers but retain an avoidant style to new objects and unfamiliar places; thus, *“a child can display an avoidant style in any of a number of contexts, but not necessarily all of them”* (p. 1483 in [13]). At the same time, a toddler with high levels of BI could be extremely reticent with adult strangers, but not necessarily so inhibited in a non-social context [14].

The general characteristics of BI tend to remain quite stable over time, although some expressions of the trait could differ according to various contexts, the child’s age, gender and cultural values [6]. According to the child’s age, the response to social and non-social stimuli may change across development: during infancy, it is more frequent to observe precursors of BI in terms of high negative reactivity, such as negative affect and high motor activity displayed by infants when confronted with novelty [15]. Conversely, during preschool and school ages, inhibited children tend to show avoidance, withdrawal, wariness, disorganization, anxiety when confronted with new stimuli [5,16].

The influence of gender also seems relevant. A meta-analysis of gender differences in temperament by Else-Quest et al. [17] indicated that girls tend to be higher in inhibitory control, shyness and some fearful behaviors, whereas boys are higher in approach-based tendencies such as anger, exuberance, and activity level. Few studies investigated specific gender differences in BI trait during toddlerhood and preschool age [18,19,20], showing that girls appeared more inhibited than boys, while other studies did not show any gender differences [21,22,23,24,25]. Gagne, Miller and Goldsmith [19] have underlined the contribution of both innate and cultural aspects to explain gender differences in the intensity of BI; however, further studies are needed to deepen this issue. 

The influence of cultural factors should also be considered, as levels of BI displayed by children of various cultures may differ [26,27,28,29,30], also according to parents’ perception [31]. In particular, a low frequency of inhibition has been observed in Western countries [26,29] and this could be in part explained by the negative connotation these cultural contexts give to shyness and inhibition, considered as socially immature and maladaptive behaviors, while assertiveness and competition are highly promoted and supported [26]. Because most of the measures for BI have been developed in Western countries, the assessment of BI should always consider the specific cultural background of the population investigated.

Following these premises, it is important to highlight the clinical relevance of studying BI; the scientific literature in this field has dedicated much attention to the observation of developmental trajectories in case of high traits of BI, showing an increased risk for anxiety disorders during childhood and adolescence [32,33,34,35,36,37,38,39,40]. Bayer and collaborators [41] have found that, among inhibited children aged 4, 47.3% of them showed anxiety disorders at 5 years and 40.2% at 6 years. Among the anxiety-related problems associated to high BI, the most frequent are higher rates of social reticence, social rejection, social anxiety and social phobia; cases of selective mutism have also been recognized [42,43,44,45,46,47]. Indeed, Clauss and Blackford [48] have reported that 40% of children displaying an inhibited profile will develop a social phobia in later childhood or adolescence, compared with the 12% of non-inhibited children.

Based on this evidence, the detection of BI from pre-school age is clinically recommended, in order to support child development and to promote timely intervention when needed. However, given the influence of child age and gender, as well as context and cultural factors, an early assessment of BI requires reliable and validated instruments to adequately detect this specific construct. 

### 1.1. The Assessment of Behavioral Inhibition

The methods for the assessment of BI most frequently used in early childhood and preschool age are laboratory assessment batteries [49], such as the Behavioral Inhibition Paradigm [50] and Laboratory Temperament Assessment Battery (LAB-TAB) [51]. These procedures consist of a series of new stimuli to which the child is exposed in a limited range of time, such as an adult stranger, unfamiliar peers, and new toys. Laboratory assessments have the advantage of collecting empirical data with a relatively low risk of rater bias [49]. Nevertheless, as no single procedure has been recognized as the gold standard for assessing BI, protocols may vary from study to study and the findings may be difficult to compare [14,52]; furthermore, the laboratory assessments are time intensive, costly, and may not fully capture how individuals act in situations outside the laboratory [53]. 

To overcome these limits, a useful method of assessment is represented by parent reports. The use of these questionnaires has indeed some advantages: they are timesaving and allow the consideration of parents’ perceptions of the child’s behaviors across a wide range of situations and longer intervals [22]; additionally, the questionnaires may collect data from different informants (e.g., mothers, fathers, teachers). 

Nevertheless, some limits also need to be acknowledged for parent-report questionnaires. There is a tendency to assess caregivers’ reports of child BI with instruments originally developed to measure other temperamental dimensions, such as shyness, fear, sociability or introversion/extroversion [6], neglecting the use of the questionnaires specifically developed for BI. Furthermore, another relevant consideration regards the moment of assessment because, as previously described, the expression of BI may vary according to child’s age [5,15,16]. For this reason, questionnaires available for infancy, such as Retrospective Infant Behavioral Inhibition Scale (RIBI) [54], could be useful for the detection of precursors, while BI is better measured during childhood. 

To our knowledge, only three questionnaires are available for the assessment during preschool age: the first two are the Preschool Behavioural Inhibition Scale (P-BIS) [55] and the Behavioural Inhibition Scale, for children aged 3–6 years (BIS) [56]; both questionnaires focus only on social contexts and neglect BI in non-social situations, therefore their use increases the risk of an overlap between the measurement of social components of BI and shyness. The third questionnaire is the Behavioral Inhibition Questionnaire (BIQ) [21], whose strength is the specific assessment of both social and non-social components of BI; therefore, it would seem a more promising measure compared to P-BIS and BIS.

### 1.2. The Behavioral Inhibition Questionnaire (BIQ) 

The BIQ was originally developed for measuring BI in children aged 3 to 5 years [20] and validated in a sample of 613 mothers, 506 fathers and 585 teachers, representative of the general Australian population. Bishop and colleagues [21] first developed a list of 40 items based on a review of literature on BI characteristics and on existing questionnaires on temperament; ten items were discarded due to poor psychometric properties and the remaining 30 items, after minor changes, constituted the final version of the BIQ. They reflected 3 specific domains: Social Novelty (3 contexts: Unfamiliar adults, Peers, Performance situations), Situational Novelty (2 contexts: Separation and preschool, Unfamiliar situations) and Physical activities with risk of injury (1 context: Physical Challenges). 

Statistical analyses found relatively robust psychometric properties both for the parent and the teacher versions. A confirmatory factor analysis supported a 6-factor structure associated to the 6 BI contexts; the authors also found an acceptable internal consistency (almost all Cronbach’s α were higher than 0.80). Moderate stability of BI was observed over the twelve-month retest period (all coefficients varied from 0.60 to 0.78 for mothers; from 0.55 to 0.74 for fathers). Moreover, a good agreement among different informants emerged: the mothers’ version significantly correlated with fathers’ (*r* = 0.69) and teachers’ versions (*r* = 0.47), and the fathers’ version with the teachers’ report (*r* = 0.43). Furthermore, concurrent and construct validity was satisfactory: a strong association (all *r*-values > 0.85) was found between BIQ total scores and the Inhibition subscale of the Temperament Assessment Battery for Children–Revised (TABCQ-R [57]); furthermore, BIQ ratings from informants and laboratory observations were significantly correlated (*r* = 0.25–0.46 for full sample and *r* = 0.35–0.60 for high- and low-BI groups). 

The validity of the BIQ was confirmed by Kim et al. [22] on an American sample of 559 parents of 3-year-old children, reporting findings consistent with the version by Bishop and colleagues [21]. They also found moderate to high convergent and discriminant validity for the parent and teacher versions, using assessment by both self-report measures (Children’s Behavior Questionnaire (CBQ) [58] and Child Social Preference Scale (CSPS) [59]) and laboratory tasks (LAB-TAB [51]).

The BIQ was validated in Dutch [4] and Israeli populations [53], confirming its good psychometric properties. Broeren and Muris [4] tested the Dutch version of BIQ (parent form) on 3 age groups of children: 4–7, 8–11 and 12–15 years old. Principal component analysis supported the 6-factor model, as originally suggested by Bishop and colleagues [21]. Overall, they found a good reliability for most of the BIQ subscales (Cronbach’s Alpha = 0.79–0.96 in 4–7-year-olds, 0.67–0.95 in 8–11-year-olds, and 0.73–0.95 in 12–15-year-olds). Further, BIQ total scores and Preschool Anxiety Scale-Revised (PAS-R) [60] were substantially correlated (all *r* > 0.66). More recently, Mernick and colleagues [53] tested the psychometric properties of a Hebrew version of the BIQ in a sample of 227 parents of 4–7-aged children. Good internal consistency (Cronbach’s Alpha = 0.94) and 3-month test-retest reliability (*r* = 0.95) were confirmed. This version also showed a good convergent validity, as BIQ total scores were significantly correlated with the Screen for Child Anxiety Related Emotional Disorders (SCARED) [61] for children aged 4–7 (*r* = 0.73) and 12–15 (*r* = 0.71); a good discriminant validity (measured with Revised Conners’ Parent Rating Scale-CPRS-R [62] ADHD index) was also found.

To date, the BIQ has been used in multiple fields of research, including investigations on neurophysiological and cognitive substrates of BI [35,63,64,65,66] and clinical studies on the risk for the onset of psychopathology [67,68], such as anxiety [36,68,69,70,71,72] and selective mutism [47]. Furthermore, the BIQ has been implemented in several studies for evaluating the efficacy of early interventions directed to inhibited or anxious children [67,73,74,75,76]. All these findings seem to provide sufficient evidence for the use of the BIQ as a reliable and valid instrument to detect child BI by assessing adults’ perceptions. 

In the light of these premises, it is relevant to further contribute to the assessment of the properties of the questionnaire, so that it can be used also in populations different from the original one [21]. Therefore, the aim of the present study was to further investigate the psychometric properties of the BIQ, giving a contribution to the validation of an Italian version of the questionnaire. Specifically, we aimed to evaluate the following psychometric properties: internal validity, internal consistency and inter-rater reliability of the Italian version of BIQ. We also assessed the convergent validity by measuring the correlation with other temperamental measures and we analyzed the discriminant validity by analyzing the divergence between BIQ and attentional difficulties. In addition, we aimed to explore the potential differences in BI traits according to children’s gender. 

Based on the results by other validation studies of the BIQ [4,22,53], we expected to also find similar good psychometric properties for the Italian version of the questionnaire. If the results would be confirmed, the study would contribute to making available the first questionnaire for the assessment of BI for Italian preschool children, as no other instruments exist, to our knowledge. Furthermore, our results could contribute to enriching the assessment of BI, adding a context belonging to Mediterranean culture, which is different from the cultural backgrounds of the populations of previous validations (Australian, American, Dutch and Hebrew).

## 2. Materials and Methods

### 2.1. Participants

Eligible participants were recruited among parents of children attending six kindergartens (18 classrooms) in Bologna and Cesena neighborhoods (Northern Italy). Kindergartens were chosen based on already ongoing collaborations with members of our research team. All parents of the children attending the six kindergartens were invited to take part to a meeting (one for each school) with the researchers, where all the information about the study was given; at the end of the meeting, participation to the study was offered. Participation in the study was voluntary and anonymous. Inclusion criteria were a good comprehension and expression of Italian language in parents, and a lack of disabilities or cognitive impairments in children. 

A total of 417 parents (230 mothers and 187 fathers) of 270 children aged from 3 to 6 years (mean age = 4.4, SD = 1.0; 47% males; 42% only child) were included in the study. The sample size was established a priori to have at least five observations for each freely estimated model parameter in CFA analysis [77]. Demographic characteristics of the sample are shown in Table 1.

Of the parents, 91% were Italian, 50% had a high school educational level, and 74% were married/cohabitant. The majority of mothers and fathers were Italian and were married. No significant differences between mothers and fathers were found for nationality (*χ*^2^(1) = 0.26; *p* = 0.62; Phi = 0.03), nor for marital status (*χ*^2^(3) = 2.43; *p* = 0.49; Phi = 0.08). With regard to the level of education, a significant difference was found between mothers and fathers (*χ*^2^(3) = 11.14; *p* < 0.05; Phi = 0.17): although the majority of mothers and fathers had an upper secondary educational level, a higher percentage of mothers had a university degree (24%) compared to the fathers (16%), and 36% of fathers had a lower secondary educational level (compared to 23% of mothers). 

To also test the convergent validity by observational measures (selected episodes from Laboratory Assessment Battery (LAB-TAB) Preschool Version), a further sub-sample of 41 parents (23 mothers, 18 fathers) of 23 children (mean age = 4.4; SD = 0.5; 42% males; 65% only child) was recruited from one of the kindergartens included in the study. Unfortunately, it was not possible to recruit a wider sample, due to some practical constraints (e.g., organization difficulties for most of the kindergartens). A priori power analysis was conducted with G*Power [78] to determine the required sample size to test the convergent and discriminant validity. We indicated the Correlation point biserial model, Alpha = 0.05, 1-Beta = 0.80, a medium effect size *r* = 0.40, and we obtained a required sample size of 44 subjects. Multilevel analysis with the school and class variables as random effects and BIQ total score as the dependent variable was applied to check for potential clustering of the data due to the different schools and different classes included in the study. Results showed no cluster effect for the school variable, considering the Intraclass Correlation Coefficient (ICC = 0.01 with Wald’s *Z* = 0.75 and *p* = 0.45), or for the class variable (ICC = 0.02 with Wald’s *Z* = 0.86 and *p* = 0.39) on BIQ total scores. 

### 2.2. Procedure and Measures

The project study was approved by the Ethics Committee of the Department of Psychology, University of Bologna, in January 2014. 

Parents who participated to the meeting with the researchers and who agreed to participate received a consent form and a set of questionnaires (BIQ, Italian Questionnaires of Temperament (QUIT) [79] and Revised Conners’ Parent Rating Scale (CPRS-R) [80]) which were handed out by their child’s teacher. All the materials were completed at home and returned to the researchers via the teacher. The presence of BI was also assessed according to observational measures (LAB-TAB) in a subsample of children.

The Behavioral Inhibition Questionnaire (BIQ) [21] consists of 30 items for the parent version, rated on a seven-point Likert scale ranging from 1 (hardly ever) to 7 (most always). Bishop et al. [21] also developed a teacher form, composed of 28 items, but for the aims of the present study we only used the parent version. The BIQ total score ranges from 30 to 180, where higher scores correspond to higher levels of Behavioral Inhibition. Items are included in three specific domains assessing children’s behaviors in multiple contexts: Social Novelty (14 items), Situational Novelty (12 items) and Physical Challenges (4 items). Social Novelty refers to the subscales: unfamiliar adults, peers and performing (in front of others), while Situational Novelty is related to the subscales: unfamiliar situations, and separation. Physical Challenges refers to novel physical activities with possible risk of injury.

For the purposes of the study, the original version of the BIQ was translated and cross-culturally adapted according to standard international guidelines as suggested by the International Test Commission [81]. In the first phase, two native Italian speakers, professionals in English translation, translated the items of the original version of the BIQ into Italian, independently. A revision panel of clinical child psychologists and experts in BI assessed the two translations and developed a first synthesized version, which was back-translated into English. Subsequently, the Italian translation, the English back-translation, and the original version were compared and synthesized in a final version. This preliminary version of the BIQ was then submitted to a gender-mixed sample of 20 parents (who were not included in the final sample of this study), in order to find minor changes to improve the readability of the instrument. After receiving the feedback from this sample of parents, the same panel of specialists discussed and approved a final Italian version of the BIQ to undergo an appropriate evaluation of its psychometric properties. The convergent validity of this parent version was evaluated by comparing the BIQ results with the QUIT [79] and LAB-TAB [51].

The QUIT [79] is represented by a sixty-item questionnaire specifically created to assess child temperament from the first month after childbirth to eleven years in the Italian population. It has been developed based on the assumptions of temperament by Thomas and Chess [82] and assesses parent’s reports (mother and/or father) on six main temperamental dimensions across four different child’s ages: 1–12 months, 13–36 months, 3–6 years (the version used for this study), 7–11 years. The six dimensions are: Physical Activity, Attention, Inhibition to Novelty, Social Orientation, Positive Emotionality, and Negative Emotionality; a high score in a specific dimension refers to a high presence of this dimension in the child’s temperament. Each item is rated on a 6-point scale ranging from 1 (hardly ever) to 6 (most always). The 3–6-year version was validated on a sample of 511 parents, showing a moderate to good internal consistency (Cronbach’s Alpha = 0.56–0.83); Pearson correlation coefficients between the scales of the questionnaire, completed separately by mothers and fathers, ranged from 0.35 to 0.74, showing a sufficient to good correlation between parents [79]. For the aims of the present study, we were particularly interested to the QUIT dimensions of Novelty Inhibition, clearly related to BI, and Negative Emotionality, because infant negative affect is a precursor of BI in childhood [12] and it is usually described as an emotional feature of BI [83].

The LAB-TAB Preschool Version [51] is a battery for the assessment of temperament dimensions and consists of 33 episodes evaluating Fear, Distress, Exuberance, Interest/Persistence, Activity Level, Inhibitory Control, and Contentment in children aged 3–6 years. In this study, children were assessed by an adapted version of LAB-TAB for the Italian cultural context, as used in previous studies [84,85]. In order to adequately respond to the aims of the study, we contacted the author of LAB-TAB (Goldsmith) to employ his supervision in the choice and implementation of LAB-TAB. For the comparison of BIQ scores with indicators of BI, the author recommended every child should undergo two episodes of the LAB-TAB procedure: the Stranger Approach (for social components of BI) and the Risk Room (for non-social aspects). The choice of these episodes for the detection of BI was also consistent with previous studies [86,87,88,89,90] and the American validation of the BIQ [22].

The Stranger Approach consists in a “simulation” of a real-life situation. A child, alone in a room, meets a stranger who tries to interact with him/her for about two minutes. The reaction of the child when the stranger approaches him/her is coded. This episode is oriented to detect the social aspects of BI. During the Risk Room episode, the child enters a room with unfamiliar or new toys; initially he/she remains alone to play (phase I), and after 5 min the researcher enters the room asking the child to approach to each toy (phase II). The reaction of the child to each unfamiliar/new object and his/her approach is then coded. This episode is oriented to detect non-social aspects of BI as identified by Kagan and colleagues [91]. Episodes were administered subsequently and only one session of observation was done for every child. All procedures were videotaped; following the LAB-TAB manual, each episode was divided into 20- or 30-s epochs, and specific variables representing the inhibited child’s typical behaviors [12,50,91] were coded for each epoch [51]. For the aims of this study, representing an initial validation of the Italian version of the BIQ, we selected specific variables to be coded. For the Stranger Approach, the variables were: (a) Intensity of decrease in activity: represents an apparent or sudden decrease in the activity level of the child during the interaction with the stranger; it includes also freezing behaviors, typical manifestations of inhibited temperament, and was rated on a 4-point scale; (b) Intensity of verbal hesitancy: refers to the hesitancy of the child’s answers to the stranger and was rated on a 3-point scale. For the Risk Room episode, the coded variables were: (a) Latency to intentionally touch first object: concerns the interval in seconds before the first definite contact with the first object; (b) Total number of objects touched: includes the number of toys (min. 1–max. 5) explored by the child during the Risk Room episode. All videos were coded by a researcher and clinical child psychologist who had expertise with BI. A second rater coded eight randomly selected videos of LAB-TAB episodes: the inter-rater reliabilities showed acceptable values, ranging from 0.69 to 0.88.

The CPRS-R [62] is a screening tool for the assessment of ADHD symptoms in children aged 3–17 years, by parents’ report. High Alpha Coefficients (range 0.75–0.95) showed excellent internal consistency of the CPRS-R. In the present study, we used the Italian version of the Conners Parents Rating Scales Revised, Short Form (CPRS-R:S) [80] that is composed of 27 items, on a 4-point Likert scale ranging from 0 (not at all true) to 3 (very much true). Eight different subscales measure: Oppositional, Cognitive Problems/Inattention, Hyperactivity, Anxious/Shy, Perfectionism, Social Problems, and Psychosomatic; additionally, it is possible to measure a global ADHD index that probes to recognize children/adolescents at risk of ADHD. Because previous literature has underlined a strong association between BI and anxiety and shyness [4,22,41,46,53], in the present study we used scores at Anxious/Shy subscale for detecting convergent validity of the BIQ. Furthermore, we used the ADHD index for the assessment of discriminant validity, as previously done in the validation of Hebrew version of the BIQ [53].

### 2.3. Statistical Analyses

The data that support the findings of this study are available on request from the corresponding author [MB]. The data are not publicly available due to restrictions (e.g., their containing information could compromise the privacy of research participants). The statistical analyses were conducted using SPSS Statistics (version 26.0; IBM, Chicago, IL, USA) and Analysis of Moment Structures (IBM SPSS Amos, version 26.0; IBM, Chicago, IL, USA) [92] software package. 

As a preliminary description of the BIQ, socio-demographic effects on behavioral inhibition were investigated by three univariate ANOVAs that were conducted on the total sample of parents, and separately on the subsamples of mothers and fathers. In all these analyses, educational level, marital status and nationality were considered as between subject factors, the age was the covariate and the BIQ total score was used as dependent variable. These preliminary analyses aimed to describe possible socio-demographic effects on behavioral inhibition. To describe the potential differences that emerged in the parents’ BIQ reports between male and female children, three multivariate ANOVAs were conducted on the total sample of parents and on the subsamples of mothers and fathers, considering child’s gender as factor and BIQ subscales and total score as dependent variables.

The structural validity of the BIQ was evaluated by means of confirmatory factor analysis (CFA). Taking into account that the data are ordinal and non-normally distributed, the unweighted least squares (ULS) estimation procedure was chosen to perform the CFA. Considering the small sample size of the mother and father subsamples and considering that acceptable sample size to perform CFA analysis requires at least five observations for each freely estimated model parameter [77], CFA analysis was conducted on the total parents’ sample (*N* = 417). The closeness of the hypothesized model to the empirical data was evaluated through the following goodness-of-fit indices: the goodness of fit index (GFI, cut-off > 0.95), the normed fit index (NFI, cut-off > 0.95), parsimony normed fit index (PNFI, cut-off > 0.50), the standardized root mean square residual (SRMR, cut-off < 0.1) [93,94,95]. To test for measurement invariance of the BIQ, we tested the model across the parental role (mothers vs. fathers).

Reliability (internal consistency) was analysed by Cronbach’s Alpha on item-total scores in total sample and in mothers and fathers, separately, and by Guttman’s split-half coefficient and composite reliability (CR) index. Inter-rater reliability was measured with inter-class correlation index and repeated measure ANOVA with the BIQ total score as dependent variable, the informant (father and mother) as within subject measure and the child’s gender as between subject factor (for this analysis we included only those couples mated on the basis of their own child).

To assess the convergent and divergent validity, Pearson’s *r* correlation analysis was applied on the BIQ total scores (separately for the parents’ sample, and for father and mother subsamples) with the subscales of the QUIT questionnaire, with the 4 LAB-TAB variables, with the Anxious-Shy score and with ADHD Index measured by CPRS-R.

Discriminant validity was assessed by and Heterotrait-Monotrait Ratio of Correlations (HTMT) index.

## 3. Results

From the ANOVAs, we found that the age, the educational level, the marital status, and nationality did not affect BIQ score both considering the total sample of parents, and separately for mothers and fathers (all *p* > 0.05).

Descriptive statistics for BIQ subscales and total scores evaluated in children by parents, and separately by mothers and fathers, are shown in Table 2.

Considering the gender differences, when the analysis was conducted on the whole sample, gender differences were found in “Peers”, “Unfamiliar adults”, “Unfamiliar situations” and in the BIQ total score (all *p* < 0.01), evidencing higher scores for females compared to males.

### 3.1. Confirmatory Factor Analysis

To evaluate the BIQ factor structure, four models for parent reports were compared. The first model (Model 1) was a single-factor model in which all items of the questionnaire were loaded onto a single dimension of inhibition. The second model (Model 2) was a three-correlated-factors model where the items of the questionnaire cluster around three domains in which BI occurs (social novelty, situational novelty, and novel physical activities involving minor risk). The third model (Model 3) was a six-correlated-factors model where the items cluster around six contexts that lay within the previous three domains (peer situations, unfamiliar adults, performing in front of others, unfamiliar situations, preschool/separation, and physical challenges). The fourth model (Model 4) was a six-correlated-factors model loading onto one higher-order-factor model, to evaluate whether the covariance between the six first-order factors could be explained by a single higher-order factor reflecting a general BI. To a descriptive purpose, and considering the possible differences deriving from the different informants, a comparison of the same four estimated models in the mother and father subsamples was performed and is available as Appendix A.

By comparing the four models, Model 4 performed optimally; therefore, we present a full description of the model in the main text, while the other models are described in Appendix A. The fit of Model 4 (six first-order factors, one second-order factor) provided a good fit to the data (Table 3), with a GFI higher than 0.95, a NFI higher than 0.95, a PNFI higher than 0.50 and an SRMR lower than 0.08. Despite that Model 3 was also a well-fitted model, and that the fit indexes were very similar between the two models, the PNFI value was slightly higher than the one associated with the first-order model. Therefore, given that the correlation among the six factors was substantial and that there is theoretical justification to consider a general factor (BI) as a higher-order construct that causes the six lower-order dimensions of BI, we concluded that the higher-order model of the BIQ questionnaire best represented the data of the Italian sample. All the BIQ items loaded significantly to the designated factor, with 27 of the 30 items loading greater than 0.40 (item 4, 14, and 17 loaded below 0.40). The loadings of the six first-order factors onto the second-order factors were also significant (Figure 1). Item 4 (“The child is cautious in activities that involve a physical challenge (e.g., climbing, jumping from a certain height”) and item 17 (“The child is hesitant to explore new play equipment”) showed the lowest loading to the designated factor (0.39 and 0.23, respectively) and belonged to the same latent variable (Physical challenges), which also was the factor showing the lower loading to the second-order factor (0.41). Additionally, item 14 (“The child is independent”) showed a low loading to the designed factor (0.26), but it belonged to the “Unfamiliar situations” latent factor that had the highest loading with the second-order factor (0.99).

Similar results were obtained performing CFAs on the mother and the father subsamples, separately (see Appendix A). However, considering the low sample size of each subsample, the results of these two additional CFAs have only a descriptive value.

### 3.2. Measurement Invariance

Configural invariance was tested across groups by requiring the same factor structure across the mother and father groups but allowing the magnitudes of all estimated parameters to vary. Metric invariance was tested across groups by setting factor loadings to be equal across the mother and father groups but allowing other estimated parameters to vary. Both the model to test configural invariance and the model to test the metric invariance showed good fit indices that did not vary significantly from one model to the other. The BIQ scale showed configural and metric invariance across groups (see Table 4).

### 3.3. Internal Consistency

BIQ total score showed an excellent internal consistency and a significant item-total correlation for the total sample of parents (Cronbach’s Alpha = 0.92; Guttman’s split-half coefficient = 0.94; item-total = 0.15–0.71). Similar results were found both for mother (Cronbach’s Alpha = 0.92; Guttman’s split-half coefficient = 0.95; item-total = 0.15–0.73) and for father (Cronbach’s Alpha = 0.90; Guttman’s split-half coefficient = 0.94; item-total = 0.13–0.67) subsamples. Only two items showed low correlation with total score (item 4 and item 17). Considering the BIQ subscales, good Alpha coefficients (ranging from 0.75 to 0.87) and good item-total correlations indexes emerged (Table 5), except for the “Physical challenges” subscale, showing a low Alpha coefficient (0.41) and low item-total correlation coefficients (ranging from 0.07 to 0.33). The item 17 had the lowest item-total correlation, showing an Alpha = 0.55 if the item was deleted. Comparable results were obtained for the mother and father forms, separately (Table 5). CR results indicated adequate (CR > 0.70) reliability for all the subscales except for the Physical Challenges subscale (Peers CR = 0.85; Physical Challenges CR = 0.48; Separation CR = 087; Performance Situations CR = 0.75; Adults CR = 0.82; Unfamiliar Situations CR = 0.81).

### 3.4. Inter-Rater Reliability

Mother and father total scores were significantly correlated (*r* = 0.63, *p* < 0.001). Furthermore, mothers and fathers showed a good inter-class correlation index (ICC = 0.77; 95% C.I. = 0.69–0.83). The repeated measures ANOVA, which considered BIQ total scores, showed a non-significant informant effect (F(1, 161) = 1.48; *p* = 0.23; Partial η^2^ = 0.01) and a non-significant informant by the child’s gender effect (F(1, 161) = 0.07; *p* = 0.79; Partial η^2^ = 0.00).

### 3.5. Convergent, Divergent and Discriminant Validity

Pearson’s correlation analysis results concerning convergent validity are shown in Table 6. Significant positive correlations emerged among BIQ total scores and “Novelty Inhibition” and “Negative Emotionality” QUIT subscales for parents, for father and mother subsamples. Additionally, significant negative correlations among BIQ total scores and “Social Orientation”, “Positive Emotionality” and “Attention” emerged, again for the main sample and the 2 subsamples. This result was confirmed by significant correlations between BIQ total scores and Anxious-Shy scores evaluated with CPRS-R in the mothers’ sample (Pearson’s *r* = 0.52; *p* < 0.01), and in the fathers’ sample (Pearson’s *r* = 0.37; *p* = 0.04).

In addition, to test the convergent validity, Pearson’s r correlation indexes between parent BIQ scores and the 4 LAB-TAB variables were calculated (Table 7). Results showed significant correlations in the parents’ sample among Intensity of Decrease Activity and BIQ total score, Separation subscale, Unfamiliar adult, and Unfamiliar situations subscales. Moreover, Total Number of Objects Touched showed significant negative correlations with Separation and Unfamiliar adult subscales, whereas Latency to intentionally touch the first object significantly correlated with Unfamiliar adult and Physical challenges subscales.

Considering only the mothers’ sample, Intensity of Decrease Activity correlated with Separation subscale, Unfamiliar adult, and Unfamiliar situations. Furthermore, Total Number of Objects Touched showed significant negative correlations with the subscales, Separation and Unfamiliar adults, while Latency to intentionally touch the first object showed a significant correlation with Unfamiliar Adult.

Globally, these findings indicated a medium magnitude of the associations between parents’ reports and observational measures of BI (both social and non-social), as all the significant correlation coefficients ranged from 0.32 to 0.52. However, when considering the fathers separately, no significant correlations were found among LAB-TAB selected variables and BIQ subscales and total score.

Divergent validity was analysed between BIQ total score and ADHD Index measured by CPRS-R, and correlations were calculated. As expected, both mothers’ and fathers’ samples showed non-significant correlations (Pearson’s *r* = 0.23; *p* = 0.19; Pearson’s *r* = −0.24; *p* = 0.19, respectively).

HTMT results indicated acceptable discriminant validity according to the HTMT 0.85 criterions (see Table 8).

## 4. Discussion

The present study aimed to evaluate the psychometric properties of the BIQ in a new cultural context, represented by a population of Italian preschool children. The findings were overall in line with other studies that investigated the operationalization of the BI construct [4,22,53] and also added new insights to former literature.

First, we found a gender effect on BIQ, showing that girls obtained higher scores than boys and confirming part of the previous literature [18,19,20,47,96]. In our study, this difference emerged particularly when children were exposed to unfamiliar peers, adults, and situations. Nevertheless, when compared to previous studies using the BIQ, our results only partially confirmed those by Bishop et al. [21], who found higher BI in girls only on the Performance Situation and Adults subscales. Furthermore, our results were not in line with the findings by Kim et al. [22] and Vreeke et al. [30], where boys were highly inhibited in Performance and Separation situations compared to girls (while these latter were more inhibited in Physical challenges), nor with those by Mernick et al. [53], who did not find any significant difference. We may consider that, despite the possible influence of methodological issues, the contrasting results reflect the debate in literature about the relationship between gender differences and BI. Globally, the theoretical perspectives on gender effects beyond the findings include, on one hand, the biological theorists, suggesting a prevalent and possible effect of factors (e.g., sex hormones in utero, which would lead to body and brain differences between males and females) existing either prenatally and/or at birth; on the other hand, other researchers suppose that differences are mainly due to gender roles and stereotypes [97]. Again, the context of observation could represent a further influencing factor [98], where girls might be inhibited especially in response to new persons (as emerged in our study), while boys in new environments. Therefore, at present, there is the need to further improve the empirical evidence to fully understand the role of child gender on BI.

Regarding the investigation of main psychometric properties of the Italian BIQ, the findings were in line with the original version and subsequent ones [4,22,53] and globally demonstrated good reliability and validity of this Italian version. CFA showed a good internal validity for the overall sample of parents: results yielded the 6-factor model loading onto a single general dimension of BI as detected by Bishop and collaborators [21] and by Kim et al. [22]. Consistently with these studies, our results seem to confirm BI as a higher-order disposition that is expressed in multiple contexts. It is also of note that, despite our values being less optimal than those resulted in other validations of the BIQ [21,22,53], the items with the lower association scores (#4, #14, #17) were the same as those which emerged also in the study by Kim et al. [22]. These results seem to be especially influenced by the scores of two of the four items that constitute the Physical Challenges subscale: #4-The child is cautious in activities that involve a physical challenge (e.g., climbing, jumping from a certain height.); and #17-The child is hesitant to explore new play equipment. A useful consideration regards the format of these items: in both cases, parents were directly asked to answer about inhibition of their children, suggesting a criticality or a negative judgement when they had to assess their children’s reticence. Conversely, more adequate values emerged for the other two items of the same scale (#13-The child is confident in activities that involve physical challenge (e.g., climbing, jumping from heights); #29-The child happily explores new play equipment), suggesting that parents are more prone to assess positive qualities, such as resourcefulness, of their children. The way the items are formulated (negative or positive), therefore, could more heavily influence the Physical Challenges subscale, given that it is composed by fewer items (four) than all the other BIQ subscales. Furthermore, the poorer fit of our model might suggest that this area is particularly critical for the evaluation of Italian children by their parents, therefore further exploration of this subscale would be recommended in future studies. 

In addition, for item 14 (“The child is independent”), some considerations are useful: differently from the other items of Unfamiliar situation subscale, this item is somehow generic and does not specify a context of application of the answer (i.e., new situations, new places); therefore, parents could be misled in their interpretation. As suggested by Gartstein [26], Mediterranean parents (i.e., Italian or Spanish populations) are used to highly supporting the preservation of close proximity, while independency is associated to a more individualistic approach, typical in other Western populations. 

Taken together, these results suggest some potential critical elements for these items to be used in the Italian context. Nevertheless, considering that our work represents a first validation of the Italian version of the BIQ, these results may be considered overall promising and future studies could further explore the role played by the abovementioned items. 

According to the internal consistency of the Italian version of the BIQ, we found excellent results given by significant item-total correlation coefficients, consistently with previous BIQ validation studies [4,22,53]. Nevertheless, low correlation values emerged in Physical Challenges subscale, as already observed by Kim et al. [22] and Mernick et al. [53]. As for CFA, our results could be influenced by item 17, the only one to show low item-total correlations in parents, and mother and father forms. Again, the specificity of the item could have contributed to its weakness for the Italian context.

Our results also indicated adequate convergent and discriminant validity of the Italian BIQ, confirming previous validation studies [4,21,22,53]. Specifically, significant correlations among BIQ total scores and those of QUIT in parents, and mother and father groups, suggested a good convergent validity. As expected, we found that BIQ total scores were significantly correlated with the QUIT Negative Emotionality and Novelty Inhibition subscales, this latter being the most relevant for the BI construct. Significant negative associations emerged also among BIQ scores and the QUIT Social Orientation, Positive Emotionality and Attention subscales. These results may confirm that inhibited children are characterized by low sociability and fewer expressions of positive emotions, such as smiles or laughter [20,22,99]; furthermore, the inverse relationship with attention seems to confirm that the construct of BI is not related to individual voluntary control [11], typical of the construct of inhibitory control. Among the 6 QUIT subscales, Motor Activity is the only one that showed no significant correlations with the BIQ total scores. Together with the weak psychometric properties observed for the Physical Challenges subscale, this result seems to suggest that the dimension of the motor area is less associated with the parental perception of BI. Prior, Broeren and Muris [4] questioned whether motor activity could sensitively reflect social aspects of BI, hypothesizing that it rather represents a measure of social aspects of anxiety (i.e., fear of body injury or physical danger). Globally, these results might highlight that the inclusion of motor area in the assessment of the BI construct needs to be better investigated and defined in future studies. 

The convergent validity of the Italian version of BIQ was confirmed by the significant correlations emerged between BIQ total score and the Anxious-Shy subscale of the CPRS-R. This result was in line with the previous versions of BIQ assessing anxiety [4,53] and shyness constructs [22]; it is also consistent with previous wider literature, where high rates of anxiety disorders were often observed in inhibited children [41,46,100].

The convergent validity was also tested considering observational scores given by LAB-TAB. Results evidenced that children characterized by a high trait of BI, rated by the overall sample of parents, showed a substantial decrease of activity (e.g., muscular tension). Specifically, this reaction emerged in the moment of separation in kindergarten (Separation subscale) or when exposed to unusual situations or strangers (Unfamiliar situation and Unfamiliar adults subscales, respectively), and in correspondence with the BIQ total score. When non-social components of BI were considered, children rated as inhibited, compared to their non-BI peers, touched fewer objects and showed a longer latency to intentionally touch the first object during physical challenges and in unfamiliar situations. These results are partially consistent with those by Kim et al. [22], even if they differently scored LAB-TAB episodes. Overall, our results on external validity are in line with previous literature comparing BI and non-BI children [12,55], showing that inhibition towards new people is associated with non-social indexes of inhibition, such as the number of objects touched by the child. Nevertheless, we did not find associations among LAB-TAB scores and Peers, Physical Challenges and Performance subscales of BIQ. This could be tied to methodological issues, as LAB-TAB being used only on a very limited sample; furthermore, despite we chose the same episodes selected by Kim et al. [22], we used a different coding and this could explain the discrepancy in the results. Therefore, these findings, even if promising, should be further confirmed.

Convergent validity also showed different results in the mother and father samples. Indeed, the results on the overall sample of parents seemed to mainly reflect the mothers’ perception. Conversely, we did not find any correlations between father BIQ scores and LAB-TAB indexes, in line with Bishop and colleagues [21], as they also found low associations for father scores regarding convergent validity. We may hypothesize some explanations for these findings: first, it might be possible that fathers, compared to mothers, less frequently observe their children in situations that are likely to trigger BI behavior [21]; they also might be more inclined to capture the child’s verbal aspects, such as initiation latency and number of prompts required, whereas mothers may be more sensitive in considering nonverbal features; furthermore, we have to keep in mind more generally that parent reports are influenced by characteristics of personality, mood states and psychopathology [101,102].

In summary, despite moderate correlations between BIQ and LAB-TAB emerging, consistent with previous literature [21,22,102,103,104,105], the joint use of self-report and observational procedures seemed to assess dimensions of BI which were not overlapping, suggesting that an integrated method of evaluation, including both observational procedures and parent-report measures, is endorsed when possible [52,106]. 

Moreover, discriminant validity was confirmed indicating sufficient separation between the BIQ subscales. 

Finally, acceptable divergent validity was found, confirming the findings by Mernick et al. [53]. Our results showed that total BIQ was not associated with restless and impulsive behavior, as measured by the ADHD Index of CPRS-R, in both the mother and father samples. 

Some limitations of the present study should be considered. First, because it was a first contribution for Italian validation, the partial limited sample size did not allow running more specific and sophisticated analyses. This was particularly evident in the case of convergent validity assessed by LAB-TAB, where only a few indexes were selected, and the sample was very restricted. Second, we focused only on parents, and the perception from other observers was not investigated; nevertheless, it could be useful to also investigate the teachers’ perception, as included in the studies by Bishop et al. [21] and Kim et al. [22]. Third, our study was only cross-sectional-based, and we did not examine the stability of the BIQ over time. In addition, the contribution and influence of specific factors like caregiver personality and psychopathology should be included. Lastly, a further validation of the BIQ’s Italian version should also explore concurrent and predictive validity. Therefore, future studies are recommended in order to confirm the psychometric characteristics of the BIQ for Italian preschool children.

## 5. Conclusions

Despite the above-mentioned limitations, to our knowledge this is the only study, up to now, to contribute to the assessment of BI in an Italian sample, and it seems to confirm the promising psychometric properties of the BIQ already uncovered from previous international literature. 

Given that the specific cultural factors of a population influence the perception and the expression of the BI trait [26,27,28,29,30], it is important to develop, through accurate validation studies, sensitive instruments for the assessment of BI. Considering the clinical implications of BI for child and adolescent mental health, the availability of a valid tool is of particular relevance for the early identification of children at higher risk for psychopathology and the realization of tailored interventions. Additionally, from a more theoretical point of view, this validation study would support Kagan and colleagues’ perspective about BI as a unitary construct which is characterized by an avoidant and fearful behavior towards both social and non-social stimuli [107].

In conclusion, this study provides further evidence of the suitability of the BIQ for the assessment of BI, as it can be easily completed by both mothers and fathers giving, as a result, a reliable measure of BI among Italian preschool children.

## Figures and Tables

**Figure 1 ijerph-18-05522-f001:**
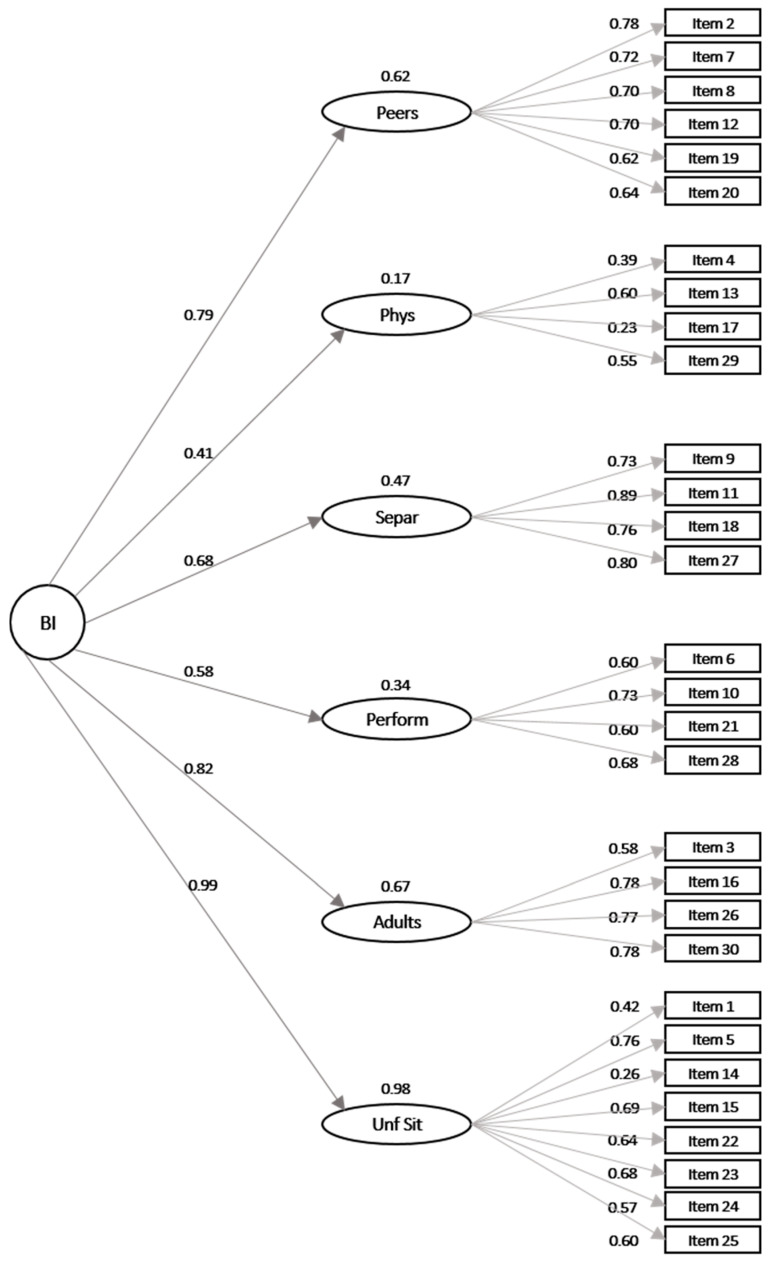
Representation of the standardized parameter estimate in Model 4. Note: BI = Behavioral Inhibition; Phys = Physical challenges; Separ = Separation; Perf = Performance situations; Unf Sit = Unfamiliar situations.

**Table 1 ijerph-18-05522-t001:** Demographic characteristics of the sample.

Sample Characteristics	Parents (*N* = 417)	Mothers (*N* = 230)	Fathers (*N* = 187)
Age, mean ± SD (range)	38.5 ± 5.4 (26–56)	37.0 ± 4.5 (26–48)	40.4 ± 5.8 (26–56)
Nationality, *n* (%)			
Italian	378 (91%)	209 (91%)	168 (91%)
Foreign	39 (9%)	20 (9%)	16 (9%)
Educational Level, *n* (%)			
Primary	6 (1%)	2 (1%)	4 (2%)
Secondary	117 (29%)	52 (23%)	65 (36%)
High school	201 (50%)	117 (52%)	84 (46%)
University	82 (20%)	54 (24%)	28 (16%)
Marital status, *n* (%)			
Married	275 (68%)	148 (66%)	127 (71%)
Living together	25 (6%)	13 (6%)	12 (7%)
Separated	16 (4%)	11 (5%)	5 (3%)
Single	86 (22%)	52 (23%)	34 (19%)

Note: Differences in sample size were due to missing data.

**Table 2 ijerph-18-05522-t002:** Estimated means and standard error (SE) for BIQ subscales and total scores assessed in parents, in mothers and in fathers, evaluating males and females.

Factors	Mean	SE	F (1384)	Mean	SE	F (1209)	Mean	SE	F (1174)
	Parents ^a^			Mothers ^b^			Fathers ^c^		
Total BIQ									
Males	89.47	2.06	11.96 **	94.52	3.00	0.09	90.99	2.89	2.24
Females	99.22	1.92	(P.η^2^ = 0.03)	95.74	2.82	(P.η^2^ < 0.01)	96.81	2.61	(P.η^2^ = 0.01)
Total	94.35	1.41		95.13	2.06		93.90	1.95	
Peers									
Males	18.20	0.59	13.15 **	18.80	0.84	4.07 *	17.73	0.82	8.03 **
Females	21.12	0.55	(P.η^2^ = 0.03)	21.13	0.79	(P.η^2^ = 0.02)	20.87	0.74	(P.η^2^ = 0.04)
Total	19.66	0.40		19.87	0.58		19.30	0.55	
Physical Challenges									
Males	10.64	0.31	1.29	10.34	0.42	1.51	10.84	0.45	0.10
Females	11.12	0.29	(P.η^2^ < 0.01)	11.05	0.40	(P.η^2^ = 0.01)	11.03	0.41	(P.η^2^ < 0.01)
Total	10.88	0.21		10.70	0.29		10.93	0.30	
Separation									
Males	12.39	0.49	4.08	14.38	0.71	2.43	12.65	0.66	0.03
Females	13.74	0.46	(P.η^2^ = 0.01)	12.87	0.67	(P.η^2^ = 0.01)	12.49	0.60	(P.η^2^ < 0.01)
Total	13.06	0.33		13.63	0.49		12.57	0.45	
Performance Situations									
Males	12.88	0.41	0.48	13.41	0.56	0.22	13.03	0.59	0.04
Females	13.26	0.38	(P.η^2^ < 0.01)	13.05	0.53	(P.η^2^ < 0.01)	12.87	0.53	(P.η^2^ < 0.01)
Total	13.07	0.28		13.23	0.39		12.95	0.40	
Adults									
Males	12.41	0.42	12.20 **	12.86	0.60	0.80	12.99	0.60	2.79
Females	14.42	0.39	(P.η^2^ = 0.03)	13.60	0.57	(P.η^2^ < 0.01)	14.34	0.54	(P.η^2^ = 0.02)
Total	13.42	0.29		13.23	0.41		13.66	0.41	
Unfamiliar situations									
Males	23.04	0.62	9.30 **	24.72	0.90	0.30	23.76	0.87	1.56
Females	25.64	0.58	(P.η^2^ = 0.02)	24.04	0.85	(P.η^2^ < 0.01)	25.22	0.78	(P.η^2^ = 0.01)
Total	24.34	0.42		24.38	0.62		24.49	0.58	

Note: ^a^ *N* = 417; ^b^ *N* = 230; ^c^ *N* = 187; ** *p* < 0.01; * *p* < 0.05; SE = Standard error; F = F test value; P.η^2^ = Partial Eta Squared.

**Table 3 ijerph-18-05522-t003:** Fit Indexes for each Model for parents, mother and father reports of Behavioral Inhibition Questionnaire (BIQ).

Model	GFI	NFI	PNFI	SRMR
Parent BIQ				
Model 1: 1 factor	0.93	0.91	0.84	0.09
Model 2: 3 correlated factors	0.95	0.92	0.85	0.08
Model 3: 6 correlated factors	0.97	0.96	0.86	0.06
Model 4: 6 first-order factors. 1 second-order factor	0.97	0.96	0.88	0.06

Note: GFI = Goodness of Fit Index; NFI = Normed Fit Index; PNFI = Parsimony Normed Fit Index; SRMR = Root Mean Square Error of Approximation.

**Table 4 ijerph-18-05522-t004:** Results of the Measurement Invariance test across parental role groups.

Model	Parental Role (Mothers and Fathers)			
	GFI	NFI	PNFI	SRMR
Configural invariance: Factor structure constrained to be equal	0.97	0.95	0.85	0.06
Metric invariance: Factor loadings constrained to be equal	0.96	0.94	0.88	0.07

**Table 5 ijerph-18-05522-t005:** Cronbach’s Alphas and Item-Total correlations for parents, and for mothers’ and fathers’ forms separately.

	Parents		Mothers		Fathers	
	Alpha	Item-Total	Alpha	Item-Total	Alpha	Item-Total
Total BIQ	0.92	0.15–0.71	0.92	0.15–0.73	0.90	0.13–0.67
Peers	0.85	0.56–0.70	0.86	0.57–0.70	0.83	0.50–0.70
Physical Challenges	0.41	0.07–0.33	0.44	0.07–0.35	0.37	0.07–0.34
Separation	0.87	0.69–0.76	0.88	0.72–0.79	0.84	0.63–0.75
Performance Situations	0.75	0.49–0.64	0.75	0.48–0.66	0.74	0.49–0.60
Adults	0.81	0.51–0.71	0.84	0.53–0.73	0.78	0.49–0.68
Unfamiliar Situations	0.80	0.27–0.68	0.83	0.26–0.70	0.76	0.27–0.69

**Table 6 ijerph-18-05522-t006:** Correlations between BIQ total scores and Italian Questionnaires of Temperament (QUIT) scores.

BIQ Total Score	QUIT Dimensions					
	Social Orientation	Novelty Inhibition	Motor Activity	Positive Emotionality	Negative Emotionality	Attention
Parents ^a^	−0.37 **	0.49 **	−0.04	−0.37 *	0.24 **	−0.16 **
Mothers ^b^	−0.37 **	0.50 **	0.01	−0.32 **	0.23 **	−0.14 *
Fathers ^c^	−0.37 **	0.47 **	−0.12	−0.45 **	0.26 **	−0.21 *

Note: ^a^
*N* = 417; ^b^
*N* = 230; ^c^ *N* = 187; ** *p* < 0.01, * *p* < 0.05.

**Table 7 ijerph-18-05522-t007:** Correlations between BIQ subscales and LAB-TAB indexes.

	LAB-TAB Social Inhibition Indexes		LAB-TAB Non-Social Inhibition Indexes	
BIQ Total Score and Subscales	Intensity of Decrease in Activity	Intensity of Verbal Hesitancy	Total Number of Objects Touched	Latency to Intentionally Touch the First Object
Parents (*N* = 41)				
Total BIQ	0.33 *	0.08	−0.25	0.25
Peers	0.28	−0.03	−0.19	0.09
Physical Challenges	−0.08	−0.04	−0.13	0.32 *
Separation	0.40 **	0.14	−0.38 *	0.21
Performance Situations	0.18	−0.16	−0.25	0.17
Adults	0.36 *	0.13	−0.39 *	0.46 **
Unfamiliar Situations	0.44 **	0.22	−0.19	−0.28
Mothers (*N* = 23)				
Total BIQ	0.39	0.13	−0.26	0.28
Peers	0.28	0.00	−0.10	0.09
Physical Challenges	−0.06	0.17	−0.22	0.39
Separation	0.52 *	0.26	−0.47 *	0.23
Performance Situations	0.10	−0.19	−0.19	0.12
Adults	0.42 *	0.27	−0.42 *	0.47 *
Unfamiliar Situations	0.43 *	0.26	−0.22	0.26
Fathers (*N* = 18)				
Total BIQ	0.34	0.02	−0.27	0.22
Peers	0.31	−0.01	−0.31	0.09
Physical Challenges	−0.02	−0.34	−0.04	0.22
Separation	0.29	−0.00	−0.26	0.20
Performance Situations	0.31	−0.11	−0.31	0.26
Adults	0.29	−0.08	−0.38	0.45
Unfamiliar Situations	0.44	0.16	−0.18	0.20

Note: ** *p* < 0.01, * *p* < 0.05.

**Table 8 ijerph-18-05522-t008:** Heterotrait-Monotrait Ratio of Correlations (HTMT) results.

BIQ Subscales	Peers	Phys	Separ	Perform	Adults
Phys	0.40				
Separ	0.54	0.31			
Perform	0.52	0.30	0.31		
Adults	0.64	0.24	0.54	0.62	
Unf Sit	0.76	0.60	0.75	0.48	0.77

Note: Phys = Physical challenges; Separ = Separation; Perf = Performance situations; Unf Sit = Unfamiliar situations.

## Data Availability

Data available on request due to privacy and ethical restrictions.

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
