# Peer review of "Validation of the Italian Version of the Behavioral Inhibition Questionnaire (BIQ) for Preschool Children"

_ijerph, 2021, doi:10.3390/ijerph18115522_

Round 1

Reviewer 1 Report

General Comment Points

This manuscript reports on an interesting topic. I applaud the aspirations represented in this paper. However, both formal and content aspects of the manuscript must be revised. I hope the suggestions I give below will support you in advancing your research efforts on this topic. Following are my specific comments on this paper.

Title Review Points

The title captures the reader’s attention but it does not clearly informs the reader about the contents of the article.

Introduction Review Points

  1. In the introduction, the important information is scarce.
  2. The introduction does not summarize, integrate, and critically evaluate the empirical knowledge on the topic of this paper.
  3. The objectives of the study should not be presented as a subsection of the introduction but integrated into the text as a response to the gap in the scientific literature.
  4. Authors should include the contributions of the study to the scientific literature.

Method Review Points

  1. Authors should describe the type of sampling used
  2. The authors should calculate the average variance extracted (AVE) to evaluate discriminant validity. Moreover, they should analyze the reliability (internal consistency) of the scales Guttman’s Split-Half Coefficient and the composite reliability index by Bagozzi and Yi (1988), too.

Discussion Review Points

  1. Authors should have subsections in the discussion
  2. The authors should improve the theoretical and practical implications of the study.

Reviewer 2 Report

After reviewing the manuscript, I think it is a very interesting and necessary article for today's society. However, there are some aspects of improvement that make it necessary, in my view, for the article to undergo revisions. The suggestions are explained below:

Introduction

- Check the quotes.

- Verify the subsections and that the importance is on the measuring instrument that is going to be evaluated.

Conclusion

- The conclusion should be better justified based on the results obtained and the previous literature.

References

- Increase the number of references in the last 10 years.

Reviewer 3 Report

The manuscript The assessment of Behavioral Inhibition in preschool children: A preliminary contribution to the Italian adaptation of the Behavioral Inhibition Questionnaire aimed to translate the BIQ into the Italian language and to validate the Italian translation by examining psychometric properties of the translated questionnaire. The manuscript is well-written and statistical methods are well-chosen. However, the fit of the 30 items of the Italian BIQ should not be considered as suitable (see comments below). Deleting some items and (carrying out additional exploratory FA) may lead to satisfactory results. In this case, the manuscript may be an important contribution the Italian literature on BI.

Introduction:

The introduction is well-written. It summarizes different aspects of behavioral inhibition and how to assess this construct. However, more information on the relationship between the BIQ and laboratory tasks or other outcome measures may be of interest. In what kind of research is the (original) BIQ currently used. What are the results?

In the section on the BIQ, please report the response scale of this instrument. Are the items rated on Likert-like scales? In this case, another estimation procedure (instead of maximum-likelihood estimation) may be more appropriate.

Materials and methods:

  • Please clarify: was only one parent (mother or father) of a child asked to participate in the study or was is possible that both parents of one child participated? In the latter case, some statistics (e.g., differences between mothers and fathers for marital status) do not make sense
  • 5, l. 221: “To test the convergent validity also by observational measures”: please specify at this point which kind of observational measures were used?
  • 5, l.236: please introduce abbreviations when used for the first time (QUIT, CPRS-R)

Statistical analyses/results:

  • This part should be more precise. For example, the description on ANOVAs p. 8 ll.344-346 is unclear. What were the dependent variables? Were these analyses done before or after the factor analyses? What was the purpose?
  • The authors report having used a maximum-likelihood estimation procedure for their CFA. The authors should check whether their data is multivariate normally distributed. If this is not the case, the authors should use a robust maximum-likelihood estimation procedure. Alternatively, for ordered data (e.g., data gathered through responses on a Likert-like response scale), other options are available: weighted least squares or diagonally weighted least squares estimation procedures.
  • In addition to the goodness-of-fit measures reported, the Tucker Lewis Index may be reported as well.
  • Please clarify (p. 10, l. 406): the authors report a RMSEA value of 0.80 for their final model, model 4. However, in the methods part, they report to use a cut-off of RMSEA <0.10 (which is actually already quite liberal, some studies use 0.07 or even 0.05). Their observed CFI value is also higher than the reported cut-off of 0.90. Moreover, some item loadings were quite low. And one item did not load on the designated factor. Therefore, taken together, the CFA should not be considered as satisfactory. Did the authors consider to remove some items from the Italian questionnaire (e.g., item 17?) and carry out an exploratory FA? This may be supported by internal consistency measures, for which the authors found low values for item 17 (and item 4) and especially low values for the physical challenges subscale.
  • In line with the comment above: did the authors collect ratings from both parents of one child? Otherwise, inter-rater reliabilities do not make any sense.

Minor:

  • authors may consider some additional proof-reading
  • 1, l. 2: there either may be a typo or parts of the sentence may be missing: “Behavioral Inhibition to Unfamiliar (BI) refers..”
